Mechanisms of action and in vivo antibacterial efficacy assessment of five novel hybrid peptides derived from Indolicidin and Ranalexin against Streptococcus pneumoniae

Jindal Hassan Mahmood 1
Zandi Keivan 2
Ong Kien Chai 3
Velayuthan Rukumani Devi 1
Rasid Sara Maisha 1
Samudi Raju Chandramathi 1
Sekaran Shamala Devi shamala@um.edu.my shamalamy@yahoo.com 1
1 Department of Medical Microbiology, Faculty of Medicine, University of Malaya , Kuala Lumpur , Malaysia
2 Department of Pediatrics, School of Medicine, Emory University , Atlanta , United States of America
3 Department of Biomedical Science, Faculty of Medicine, University Malaya , Malaysia
Flores-Valdez Mario Alberto
Electronic publication date: 2017 Oct 5
Publication date: 2017
Volume: 5
Electronic Location ID: e3887
Received 2017 Jun 2; Accepted 2017 Sep 13
Copyright: ©2017 Jindal et al.
Copyright year: 2017
Copyright holder: Jindal et al.
License: This is an open access article distributed under the terms of the Creative Commons Attribution License, which permits unrestricted use, distribution, reproduction and adaptation in any medium and for any purpose provided that it is properly attributed. For attribution, the original author(s), title, publication source (PeerJ) and either DOI or URL of the article must be cited.
License URL: https://creativecommons.org/licenses/by/4.0/

Keywords: Antimicrobial peptides, Hybrid peptides, Streptococcus pneumoniae, Antibiotic resistance, Syngergistic effects

Funding: University of Malaya High Impact Research UM.C/HIR/MOHE/MED/40 H-848 20001-E000079 University of Malaya Postgraduate Research Fund (PPP) PG090-2016A This study was supported by University of Malaya High Impact Research Grant (reference number: UM.C/HIR/MOHE/MED/40, account number: H-848 20001-E000079) and University of Malaya Postgraduate Research Fund (PPP) (Project no. PG090-2016A). The funders had no role in study design, data collection and analysis, decision to publish, or preparation of the manuscript.

==============================
Background

Antimicrobial peptides (AMPs) are of great potential as novel antibiotics for the treatment of broad spectrum of pathogenic microorganisms including resistant bacteria. In this study, the mechanisms of action and the therapeutic efficacy of the hybrid peptides were examined.

Methods

TEM, SEM and ATP efflux assay were used to evaluate the effect of hybrid peptides on the integrity of the pneumococcal cell wall/membrane. DNA retardation assay was assessed to measure the impact of hybrid peptides on the migration of genomic DNA through the agarose gel. In vitro synergistic effect was checked using the chequerboard assay. ICR male mice were used to evaluate the in vivo toxicity and antibacterial activity of the hybrid peptides in a standalone form and in combination with ceftriaxone.

Results

The results obtained from TEM and SEM indicated that the hybrid peptides caused significant morphological alterations in Streptococcus pneumoniae and disrupting the integrity of the cell wall/membrane. The rapid release of ATP from pneumococcal cells after one hour of incubation proposing that the antibacterial action for the hybrid peptides is based on membrane permeabilization and damage. The DNA retardation assay revealed that at 62.5 µg/ml all the hybrid peptides were capable of binding and preventing the pneumococcal genomic DNA from migrating through the agarose gel. In vitro synergy was observed when pneumococcal cells treated with combinations of hybrid peptides with each other and with conventional drugs erythromycin and ceftriaxone. The in vivo therapeutic efficacy results revealed that the hybrid peptide RN7-IN8 at 20 mg/kg could improve the survival rate of pneumococcal bacteremia infected mice, as 50% of the infected mice survived up to seven days post-infection. In vivo antibacterial efficacy of the hybrid peptide RN7-IN8 was signficantly improved when combined with the standard antibiotic ceftriaxone at (20 mg/kg + 20 mg/kg) as 100% of the infected mice survived up to seven days post-infection.

Discussion

Our results suggest that attacking and breaching the cell wall/membrane is most probably the principal mechanism for the hybrid peptides. In addition, the hybrid peptides could possess another mechanism of action by inhibiting intracellular functions such as DNA synthesis. AMPs could play a great role in combating antibiotic resistance as they can reduce the therapeutic concentrations of standard drugs.

Introduction

Pneumococcus is a major human respiratory pathogen in both children and adults (Jacobs, 2004; Cao et al., 2007). This bacterial pathogen is capable of causing both invasive and non-invasive diseases (Moschioni et al., 2012). Globally, this pathogen is responsible for 1.6 million deaths each year, of which 0.7 to 1 million are children below five years old, especially in Asian and African countries (O’Brien et al., 2009; Bravo, 2009). Since the last three decades, there has been an enormous increase in the incidence of antibiotic-resistant pneumococci, due to the extensive use of inappropriate antimicrobials (Zhou et al., 2012). Although Pneumococcal conjugate vaccine PCV served as a great tool against antibiotic resistance by S. pneumoniae and helped reduce the frequency of vaccine serotypes, there has been a considerable rise in the disease induced by non-vaccine serotypes (Reynolds et al., 2014). Hence, newer classes of antibacterial agents to overcome this serious issue are a top priority worldwide.

One of the advantageous alternatives to today’s antibiotics is antimicrobial peptides (AMPs) (Deslouches et al., 2013; Sánchez-Vásquez et al., 2013). AMPs are synthesized by almost all living beings as the first line of defense in their immune system against microbial infection. Many aspects favor AMPs over traditional antibiotics: a broad range of antimicrobial activity against pathogenic micro-organisms (including viruses, parasites, bacteria, and fungi), microbial pathogens are less efficient in developing resistance against AMPs, as killing take place in a short contact time and AMPs can act in a synergistic manner with traditional antibiotics (Yeaman & Yount, 2003; Torcato et al., 2013; Xi et al., 2013). In our earlier study, we had designed 13 novel antimicrobial peptides based on two naturally occurring templates, indolicidin and ranalexin (Jindal et al., 2015). Of these, five hybrid peptides (RN7-IN10, RN7-IN9, RN7-IN8, RN7-IN7 and RN7-IN6) presented the most potent antimicrobial activity against 30 pneumococcal clinical isolates. The MICs of RN7-IN10, RN7-IN9, RN7-IN8 and RN7-IN6 ranged from 7.81 to 15.62 µg/ml for each peptide, while the MIC of RN7-IN7 was 62.5 µg/ml. Also, none of the hybrid peptides revealed any cytotoxic effects against human cells at their MIC levels. RN7-IN10 peptide was designed by fusing the first seven amino acids at the N-terminus (FLGGLIK) of ranalexin with the 4th to 13th residual fragment (WKWPWWPWRR) of indolicidin. Likewise, RN7-IN9, RN7-IN8, RN7-IN7 and RN7-IN6 were also designed by trimming the first seven amino acid residues of ranalexin and fusing it with 5th to 13th, 6th to 13th, 7th to 13th and 8th to 13th residual fragments of indolicidin (Jindal et al., 2015) in order to preserve the biological activity of both segments in the newly designed hybrid peptides (Table 1 illustrates the sequences and physicochemical properties of all five hybrid peptides). In this study, we describe the mechanisms of action, the in vitro synergism effect and the in vivo antibacterial efficacy of the hybrid peptides against Streptococcus pneumoniae.

Table 1 Sequences and physicochemical properties of the template and hybrid AMPs.

Peptide	Sequence	MICa	aab	MWc	Qd	Pho%e	
Indolicidin	ILPWKWPWWPWRR-NH2	15.62–31.25	13	1907.30	+4	53%	
Ranalexin	FLGGLIKIVPAMICAVTKKC-OH	62.5	20	2105.70	+3	65%	
RN7-IN10	FLGGLIKWKWPWWPWRR-NH2	7.81–15.62	17	2300.791	+5	52%	
RN7-IN9	FLGGLIKKWPWWPWRR-NH2	7.81–15.62	16	2114.578	+5	50%	
RN7-IN8	FLGGLIKWPWWPWRR-NH2	7.81–15.62	15	1986.408	+4	53%	
RN7-IN7	FLGGLIKPWWPWRR-NH2	62.5	14	1800.195	+4	50%	
RN7-IN6	FLGGLIKWWPWRR-NH2	7.81–15.62	13	1709.078	+4	53%	
Notes.

a Minimum inhibitory concentration (µg/ml).

b Number of amino acids.

c Molecular weight.

d Net charge. Lys (K), Arg (R), and C-terminal amidation (NH2) was assigned with +1 charge.

e Hydrophobic residues%.

Materials and Methods

Bacterial culture and assay medium

S. pneumoniae clinical strains used in this study were obtained from University of Malaya Medical Centre (UMMC). Columbia agar with 5% sheep blood was used to culture the bacteria. Mueller-Hinton broth (MHB) was used for synergism assay and cationally adjusted as described in the guidelines of Clinical and Laboratory Standards Institute (2012).

Transmission electron microscopy (TEM)

Bacteria were prepared for TEM according to the guidelines of the Electron Microscopy Unit at the Faculty of Medicine, University of Malaya. S. pneumoniae cultures were grown overnight on Columbia agar with 5% sheep blood and suspended in cationally adjusted Mueller-Hinton broth (CAMHB) at 108 CFU/ml. Pneumococcal suspensions were incubated with hybrid peptides RN7-IN10, RN7-IN9, RN7-IN8 and RN7-IN6 at 125 µg/ml and with RN7-IN7 at 500 µg/ml (8 × of their respective MIC) in a 1.5 ml eppendorf tube for 1 h at 37 °C under 5% CO2. Cells in cationally adjusted Mueller-Hinton broth (CAMHB) were used as an untreated control. After an hour of incubation, the pneumococcal cells were centrifuged to discard the medium, washed thrice with 10 mM phosphate buffer saline at pH 7.3 and fixed overnight in 4% (v/v) glutaraldehyde. All samples were washed twice with cacodylate buffer, incubated for 2 h in osmium tetroxide buffer (OsO4 1: 1 cacodylate), washed twice with cacodylate buffer and then incubated overnight in cacodylate buffer. All the samples were washed with distilled water twice and incubated for 10 min with uranyl acetate. After this, all samples were then washed twice with distilled water and dehydrated in an ascending series of ethanol: 35% (10 min), 50% (10 min), 70% (10 min), 95% (15 min), and thrice in 100% ethanol (15 min). After dehydration, samples were incubated with propylene oxide (15 min), propylene oxide 1:1 Epon (1 h), propylene oxide 1:3 Epon (2 h) and finally incubated overnight with Epon. All the samples were embedded in agar 100 resin at 37 °C for 5 hr and maintained at 60 °C until viewing. Reichert Ultramicrotome copper grids 3.05 mm (300 square mesh) (Agar Scientific, Essex, United Kingdom) were used to prepare Ultrathin sections. Ethanol-based uranyl acetate and lead citrate were used to stain the samples for 5 min. Transmission electron microscope (Leo Libra 120) was used to capture the images.

Scanning electron microscopy (SEM)

S. pneumoniae with a starting inoculum of 1 × 108 CFU/ml in CAMHB medium, was treated with hybrid peptides RN7-IN10, RN7-IN9, RN7-IN8 and RN7-IN6 at 125 µg/ml and with RN7-IN7 at 500 µg/ml of respective peptides and incubated for 1 h at 37 °C for under 5% CO2. After incubation, 20 µl of the untreated and treated cell suspensions were transferred onto membrane filters and processed as described by the standard guidelines provided by Electron Microscopy Unit, Faculty of Medicine, University of Malaya. Briefly, the bacterial samples were fixed overnight with 4% glutaraldehyde at 4 °C and then washed twice with sodium cacodylate buffer for 10 min each. In the second fixation, 1% osmium tetroxide was used to fix the samples for 1 h at 4 °C and then washed twice with distilled water for 10 min each. All the samples were then dehydrated through a serially graded ethanol (30%, 50%, 70%, 80%, 90%, 95% and twice in 100%) for 15 min each, followed by dehydration in ethanol:acetone mixtures (3:1, 1:1 and 1:3) for 15 min each and three rounds of pure acetone for 20 min each. The samples were then dried for 1 h and kept in a desiccator before the examination. The samples were then mounted on stubs, coated with gold in sputter coater and viewed under the FEI-Quanta 650 Scanning Electron Microscope.

ATP efflux assay

The amount of ATP released from pneumococcal cells incubated with hybrid peptides was measured as described previously, with a slight modification (Tanida et al., 2006). The ATP determination kit (Molecular Probes, Eugene, OR, USA) was used to measure the amount of ATP released based on the luciferin/luciferase method according to the manufacturer’s instructions. Briefly, S. pneumoniae were grown overnight on Columbia agar with 5% sheep blood. Bacterial suspensions were spectrophotometrically adjusted to (1 × 107 CFU/ml) and incubated with hybrid peptides RN7-IN10, RN7-IN9, RN7-IN8 and RN7-IN6 at 125 µg/ml and with RN7-IN7 at 500 µg/ml. The amount of ATP released form the pneumococcal cells were measured at three time points 1, 2 and 3 h. The samples were then centrifuged at 5,000 rpm for 5 min, and ATP efflux was subsequently estimated using an ATP standard curve. Values were obtained from three independent experiments. Ceftriaxone and erythromycin were used as positive controls.

Gel retardation assay

This assay was carried out as described previously, with minor modifications (Li et al., 2013). S. pneumoniae were grown overnight on Columbia agar with 5% sheep blood. A few bacterial colonies were transferred into a 1.5 ml eppendorf tube containing phosphate buffered saline (PBS). The bacterial cells were centrifuged, PBS was discarded and 50 µl of TE buffer containing 0.08 g/ml of lysozyme and 150 U/ml of mutanolysin was added to the cells. Genomic DNA was isolated from pneumococcal cells using DNeasy Blood & Tissue Kit (Qiagen, Hilden, Germany), following the manufacturer’s giudlines. The optical density ratio of 260 nm and 280 nm (OD260∕OD280 = 1.83) was used to measure the purity of the DNA. Genomic DNA (250 ng) was incubated with hybrid peptides at various concentrations (0.24–500 µg/ml) in 12 μl at room temperature for 10 min. 2 μl of loading buffer were added to the mixture and the migration of DNA through 1% agarose gel was evaluated by electrophoresis in 1 × Tris borate–EDTA buffer (45 mM Tris–borate and 1 mM EDTA at pH 8.0) and spotted by the fluorescence of gel stain (Gel Red, BIOTIUM, Fremont, CA, USA). The natural peptide indolicidin was used as positive controls, ceftriaxone and erythromycin were used as negative controls.

Synergistic effect

Pneumococcal cells were cultured overnight using Columbia agar with 5% sheep blood at 37 °C under 5% CO2, resuspended in cation-supplemented Mueller-Hinton broth and adjusted to 5 ×105 CFU/ml, following CLSI guidelines. Combinations of hybrid AMPs with each other and with standard drugs (ceftriaxone and erythromycin) were assessed for their synergistic effects by the checkerboard titration method described previously, with minor modification (Bajaksouzian et al., 1996). Briefly, 50 µl of eight serial two-fold dilutions of drug B starting at 4 × MIC were added to each column of the plate followed by 50 µl of a fixed 0.25 × MIC of drug A, this yielded eight peptide-peptide combinations at different ratios. 100 µl of bacterial suspension (5 ×105 CFU/ml) were then added to each well and the plates were incubated for 24 h at 37 °C under 5% CO2. The fractional inhibitory concentration (FIC) index of each combination was calculated according to the following formula: FICI=MIC of drug A in combinationMIC of drug A alone+MIC of drug B in combinationMIC of drug B alone.

MIC A in combination and MIC B in combination represent the MICs of drug A and B tested in combination. MIC A alone and MIC B alone represent the MICs of drug A and B in standalone. FIC index values were interpreted as follows: two drugs have synergy if FIC ≤ 0.5, additive or indifference if 0.5 < FIC ≤ 4.0 and antagonism if FIC > 4.0. The experiment was done in triplicate.

In vivo assessment

Mice and environmental conditions

In this study, 4-week old male, pathogen free ICR (CD-1) mice were purchased from InVivos (Lim Chu Kang, Singapore) and used to assess the in vivo toxicity and antibacterial efficacy of the novel peptides, as these animals are by far the most commonly used model for the study of pneumococcal disease (Chiavolini, Pozzi & Ricci, 2008). The mice were kept in ventilated polycarbonate cages (12 h light/dark cycle, 20 ± 2 °C and 55% relative humidity). All mice were familiarized for seven days before any experimental procedure and were given unlimited pellets and water ad libitum. All animal experimentations were conducted according to the guidelines approved by Faculty of Medicine Institutional Animal Care and Use Committee (FOM IACUC), University of Malaya (ethics Reference no. : 2013-07-15/MMBTR/SDS).

In vivo toxicity

In order to evaluate the possible toxic effects correlated to peptides administered in mice, four hybrid peptides RN7-IN10, RN7-IN9, RN7-IN8 and RN7-IN6 were chosen for in vivo toxicity assessment, due to their promising in vitro antibacterial activity (Jindal et al., 2015). Mice were separated into four groups (each with four mice) and were injected with respective peptides at 1 h, 12 h, and 24 h (three-dose regimen) via IP, SC, and IN administration routes. The hybrid peptides were first administered at high doses (100 mg/kg for IP route, 100 mg/kg for SC routes and 20 mg/kg for IN route). Any abnormal behavior was recorded and survival of mice was noted as well. In case adverse effects such as high physical stress, severe lethargy, physical inactiveness, and/or death were detected, lower graded doses were given. All the administered mice were monitored for seven days or until death occurred. At day seven post administration, all animals were sacrificed and blood and organs were collected. Untreated mice were used as a control group.

In vivo antipneumococcal activity

Two pneumococcal infection models developed previously in our lab (Le et al., 2015) were used to assess the therapeutic efficacy of peptides in vivo. The systemic infection model was used to mimic pneumococcal bacteremia in humans and the pneumococcal pneumonia model was used to mimic pneumococcal pneumonia in humans. A highly virulent strain was used in both the models. The bacterial isolate was grown overnight on Columbia agar with 5% sheep blood at 37 °C under 5% CO2. The bacterial suspension was adjusted to OD625 0.08-0.1 (1 ∼ 2 × 108 CFU/ml). Mice tested for lethal systemic infection were inoculated with 1.5 × 102 CFU/mouse (100 μl) via IP route. Mice used to assess the Pneumococcal pneumonia model were inoculated with pneumococcal cells of 5 × 103 CFU/mouse (50 μl) via the intrathoracic route. Both the infection models caused 100% death within 2 to 4 days post-infection.

After 1 hr of inoculation, Mice receiving treatment were randomized and divided into six groups. RN7-IN10 and RN7-IN8 were tested at three different doses for each (5 mg/kg, 10 mg/kg and 20 mg/kg) using a group of 10 mice. Graded doses of ceftriaxone (5 mg/kg, 10 mg/kg, 20 gm/kg, 40 mg/kg and 80 mg/kg) were also tested to assess the in vivo antibacterial activity of this antibiotic. Only mice injected with PBS were served as uninfected control. Mice injected with bacterial inoculum were used as untreated control group and given sterile distilled water only. Survival of mice was documented for seven days or until death. After seven days, the experiment was ended, the blood and homogenates of the five major organs (kidney, brain, spleen, liver and lung) of the surviving mice were plated on Columbia agar with 5% sheep blood, to detect the presence of pneumococcal cells.

In vivo synergy assessment of peptide/peptide and peptide/ceftriaxone

After evaluating the in vivo antibacterial activity of the hybrids in the standalone mode, the in vivo efficiency of the hybrid peptides in combination with each other and with the standard antibiotic ceftriaxone was carried out. Graded doses of peptide and ceftriaxone were chosen and prepared at 2 × the desired concentration separately in 1 ml tubes, and the volume was 0.1 ml. Just before injection, both the drugs were combined, giving the final desired concentration at a volume of 0.2 ml. The synergetic effect was then performed in infection models (n = 10).

Anesthesia and necropsy

Mice used to evaluate the in vivo toxicity and antibacterial activity of the hybrid peptides using the subcutaneous (SC) and intranasal (IN) administration routes, were anesthetized using a combination of a standard dose of xylazine (ilium xylazil-20, 10 mg/kg) and ketamine (Narketan®-10, 100 mg/kg) through intraperitoneal (IP) injection. After seven days of treatment, the in vivo toxicity and antibacterial efficacy experiments were ended and the surviving mice were anesthetized. Blood samples for Hematological and biochemical analysis were collected via cardiac puncture using a 25G syringe (BD bioscience, San Jose, CA, USA). Whole blood for heamatological analysis was collected in 500 µl dipotassium EDTA microtainer tubes (BD Bioscience, USA). About 500 µl of blood collected an eppedorff tube and centrifuged at 8000 rpm for 5 min and then the serum was transferred into a new 1.5 ml tube for biochemistry analysis. The mice were then euthanized by cervical dislocation, dissected and the following organs were collected for histopathology evaluation: lung, kidney, brain, liver and spleen.

Hematological and biochemical analysis

For whole blood analysis, the parameters were number of red cells (RBC), number of white cells (WBC), lymphocytes, monocytes, eosinophil, granulocytes, haemoglobin (Hgb), mean corpuscular volume (MCV), hematocrit (HCT), platelet Counts (PLT), Mean corpuscular haemoglobin (MCH) and corpuscular haemoglobin concentration (MCHC). For biochemistry analysis, the parameters were alanine transaminase (ALT), creatinine, alkaline phosphatase (ALP), aspartate aminotransferase (AST), total bilirubin and urea.

Histopathological examination

The following organs were collected from all the dissected mice: lung, kidney, brain, spleen, and liver. All tissues were fixed in 10% (v/v) buffered formalin and processed for paraffin embedding. Hematoxylin–eosin (HE) was used to stain the histological sections at the histopathology laboratory, Veterinary Laboratory Service Unit, University Putra Malaysia (UPM).

Statistical analysis

GraphPad Prism 5 was used to perform the Statistical analysis. The results were expressed as mean ± standard deviation. Two-way ANOVA with Bonferroni post-test was used to analyze the significance of the difference between the treated groups and control in ATP assay. One-way ANOVA with post-hoc Dunnett-t test was used to assess the statistical difference between the blood haematogram and blood serum biochemistry parameters of the treated and the untreated control groups in the in vivo toxicity assay. Kaplan–Meier analysis with log-rank test (Mantel-Cox) was used to generate the survival curve for each treated group versus untreated control, for both in vivo antibacterial activity and in vivo synergy assays.

Results

Effects of hybrid peptides on cell morphology and membrane permeability

TEM and SEM studies were performed to observe the damaging effect of the hybrid peptides on the pneumococcal cell wall/membrane. The images obtained clearly indicated that all the hybrid peptides were capable of disrupting the integrity of bacterial membranes. As shown in Fig. 1A, the untreated cells appeared with complete cell wall and plasma membrane and therefore preserved the normal integral shape of S. pneumoniae. The pneumococcal capsular polysaccharide appeared as a thin layer sheltering the whole cell and the cytoplasm of the cell was compactly packed and occupied the entire space (Fig. 1A, arrow 1). Incubation of pneumococcal cells with hybrid peptides had led to a dramatic effect on the morphology of bacterial surface. After 1 h of incubation, the hybrid peptides were able to breach the intactness of the cell wall and/or plasma membrane, causing membrane breakage and loss of fragments (Figs. 1B–1F, arrow 2). Additionally, our TEM results have revealed that treatment with hybrid peptides has led to the leakage of the cytoplasmic components to the outer environment through the disruption of the cell wall. As a result, huge halos were detected in the inner space of all these treated cells, leading to cell collapse and death (Figs. 1B–1F, arrow 3). Moreover, the TEM results also revealed partial disconnection of the cell wall from the cell membrane in pneumococci treated with hybrid peptides, especially those treated with RN7-IN9, RN7-IN8, RN7-IN7 and RN7-IN6 (Figs. 1C–1F, arrow 4).

Figure 1 Transmission electron micrographs of S. pneumoniae after treatment with hybrid peptides.

(A) Control cells without treatment appeared with normal shape (A, arrow 1). (B–F) display the damage of pneumococcal cells after 1 hr incubation in presence of (B) RN7-IN10, (C) RN7-IN9, (D) RN7-IN8, (E) RN7-IN7 and (F) RN7-IN6. (Arrow 2) Breakage and loss of cell wall/membrane fragments. (Arrow 3) Leakage of cytoplasm and halos formation. (Arrow 4) Detachment of cytoplasmic membrane from pneumococcal cell wall. Bar indicates 200 nm.

Figure 2 Scanning electron micrographs of S. pneumoniae after treatment with hybrid peptides.

(A) Control cells without treatment appeared with normal shape and smooth surface (arrow 1). (B–F) show the severe morphological changes and surface disruption (arrow 2) of pneumococcal cells following 1 hr incubation in presence of (B) RN7-IN10, (C) RN7-IN9, (D) RN7-IN8, (E) RN7-IN7, and (F) RN7-IN6. Bar indicates 2 µm.

Scanning electron microscopy (SEM) was employed to understand the impact of the hybrid peptides on the morphology of S. pneumoniae. As presented in Fig. 2, the hybrid peptides were able to induce significant morphological alterations to pneumococcal cells. The untreated S. pneumoniae displayed normal and smooth surface (Fig. 2A, arrow 1), whereas S. pneumoniae treated with hybrid peptides at 8 × MIC appeared with a rough and injured surface (Figs. 2B–2F, arrow 2). The numerous fragments observed on the bacterial surface are an indication of cell wall breakage and fragments loss upon treatment with hybrid peptides. This result indicates that hybrid peptides could disrupt and damage the integrity of cell wall/membrane or breach the membrane, which was in agreement with the result of TEM.

In order to evaluate the permeability of the membrane and leakage of intracellular components upon treatment with hybrid peptides, the level of ATP in the supernatant following contact of the pneumococcal cells with hybrid peptides was determined using the ATP release assay. After 1 h of treatment with hybrid peptide, the levels of ATP released after 1 h of treatment with RN7-IN10 and RN7-IN9 were the highest among the five hybrid peptides tested (54.5 ± 4.7 and 42.27 ± 92 pM respectively) (Fig. 3). The ATP efflux steadily decreased, and the ATP release reached 25.42 ± 3.51 pM and 22.9 ± 3.22 pM after 3 h of treatment with RN7-IN10 and RN7-IN9 (Fig. 3). On the other hand, the quantities of ATP released from pneumococcal cells upon incubation with RN7-IN8, RN7-IN7 and RN7-IN6 after 1 h were 39.03 ± 0.2 pM, 22.35 ± 0.9 pM and 14.8 ± 0.35 pM. The levels of ATP release from pneumococcal cells treated with RN7-IN8, RN7-IN7 and RN7-IN6 were 20.54 ± 1.03 pM, 13.02 ± 2.26 pM and 11.47 ± 0.32 pM respectively after 3 h of treatment (Fig. 3). However, all the hybrid peptides showed better capacity in efflux ATP from pneumococcal cells in comparison with standard antibacterial drugs ceftriaxone and erythromycin. The efflux levels of ATP by ceftriaxone and erythromycin treated cells after 1 h of incubation were 5.24 ± 1.43 pM and 0.49 ± 0.004 pM, respectively (Fig. 3).

Figure 3 The influence of peptides on ATP release.

Two-way ANOVA with Bonferroni post-test was used to perform the statistical analysis. An asterisk (*) adjacent to peptide name directs statistical significance (P < 0.001). (A) shows the amount of ATP released upon treatment with RN7-IN10, RN7-IN9 and RN7-IN8. (B) shows the amount of ATP released upon treatment with RN7-IN7 and RN7-IN6. The experiment was done in triplicate. All Hybrid peptides presented stronger ATP efflux activity (P < 0.001) in comparison with the standard drugs erythromycin (P > 0.05) and ceftriaxone (P > 0.05).

DNA retardation activity

To clarify the influence of the hybrid peptides on pneumococcal genomic DNA, the retardation of DNA by the hybrid peptides at various concentrations was assessed by analyzing electrophoretic movement of pneumococci DNA bands through the agarose gel (1%, w/v). Our results clearly indicated that like their parent peptide indolicidin, all the five hybrid peptides were capable of inhibiting DNA migration through the gel at a concentration of 62.5 µg/ml (Figs. 4A–4F). On the other hand, the standard drugs ceftriaxone and erythromycin could not prevent the migration of DNA band through the agarose gel up to a concentration of 500 µg/ml (Figs. 4G and 4H).

Figure 4 The influence of the hybrid peptides on the migration of S. pneumoniae genomic DNA through the agarose gel.

Genomic DNA (250 ng) was mixed with peptides at different concentrations (0.24–500 µg) were mixed at room temperature for 10 min in TE buffer, and the reaction mixtures were applied to a 1% agarose gel electrophoresis. The natural peptide indolicidin was used as positive control, erythromycin and ceftriaxone were used as negative controls. All the hybrid peptides and their parent indolicidin were able to inhibit the migration of DNA at 62.5 µg/ml.

In vitro synergistic effects of peptide/peptide and peptide/antibiotic combinations

The in vitro antibacterial activity of peptide/peptide and peptide/antibiotic combinations was evaluated using the chequerboard dilution assay. Our results reveal that combinations of hybrid peptides (RN7-IN10, RN7-IN9, RN7-IN8, RN7-IN7, and RN7-IN6) with each other showed synergistic effects, with FICI of less than 0.5 (Table 2), regardless of the susceptibility of pneumococcal isolates towards standard drugs. Likewise, combinations of standard drugs ceftriaxone and erythromycin with all five hybrid peptides presented synergistic effects with fractional inhibitory concentration (FIC) index of ≤ 0.5 against both isolates of S. pneumoniae, regardless of their susceptibility to antibiotics (Table 2). These results indicate that all the hybrid peptides were able to enhance the antibacterial activity of both the standard drugs ceftriaxone and erythromycin.

Table 2 FIC index of various combinations of hybrid peptides with each other and with standard antibiotics against susceptible and resistant S. pneumoniae.

Combination	Susceptible S. pneumoniae	Resistant S. pneumoniae	
Drug A	Drug B	FIC indexa	Interpretation	FIC indexa	Interpretation	
RN7-IN10	RN7-IN9	0.50	Synergism	0.37	Synergism	
RN7-IN8	0.28	Synergism	0.28	Synergism	
RN7-IN7	0.37	Synergism	0.50	Synergism	
RN7-IN6	0.26	Synergism	0.31	Synergism	
Ceftriaxone	0.37	Synergism	0.31	Synergism	
Erythromycin	0.26	Synergism	0.28	Synergism	
RN7-IN9	RN7-IN8	0.37	Synergism	0.50	Synergism	
RN7-IN7	0.50	Synergism	0.50	Synergism	
RN7-IN6	0.28	Synergism	0.37	Synergism	
Ceftriaxone	0.31	Synergism	0.37	Synergism	
Erythromycin	0.28	Synergism	0.26	Synergism	
RN7-IN8	RN7-IN7	0.50	Synergism	0.50	Synergism	
RN7-IN6	0.31	Synergism	0.37	Synergism	
Ceftriaxone	0.31	Synergism	0.37	Synergism	
Erythromycin	0.28	Synergism	0.26	Synergism	
RN7-IN7	RN7-IN6	0.50	Synergism	0.50	Synergism	
Ceftriaxone	0.50	Synergism	0.50	Synergism	
Erythromycin	0.37	Synergism	0.37	Synergism	
RN7-IN6	Ceftriaxone	0.37	Synergism	0.31	Synergism	
Erythromycin	0.28	Synergism	0.28	Synergism	
Notes.

a FIC index ≤ 0.5 represents synergy; >0.5 – ≤4.0 represents indifference; >4.0 represents antagonism.

Highlighted in bold: peptide-antibiotic combination with synergistic effect.

In vivo toxicity of hybrid peptides

The in vivo toxicity of four hybrid peptides namely RN7-IN10, RN7-IN9, RN7-IN8 and RN7-IN6 was evaluated following a three dose regimen with the mice at 1 h, 12 h and 24 h using three different administration routes. The results revealed that in the case of mice treated with all hybrid peptides via subcutaneous (SC) injection at the maximum dose (100 mg/kg), no animal death or hypersensitivity reactions were observed up to seven days post-treatment. Minor differences were noted for mice given RN7-IN10 via SC, which displayed significantly lower granulocytes (p = 0.0172) and ALP (p = 0.0037) (Table S1, highlighted in yellow), while Mice treated with RN7-IN6 displayed significantly higher platelet counts (p = 0.0487) in comparison with the control group (Table S1, highlighted in blue). Histopathological studies were performed with the lung, brain, liver, spleen and kidney of control and treated animals. No histological abnormalities were detected in the organs of any group, as all the tissue sections were normal and did not display differences with the control group (Fig. S1). Similarly, no abnormal physical behavior was noted upon giving the mice hybrid peptides via the intranasal (IN) route. However, treatment with RN7-IN9 displayed significantly lower MCV (p = 0.001) (Table S2, highlighted in yellow) than the control group. Mice treated with RN7-IN6 displayed significantly lower percentage of granulocytes (p = 0.0482) (Table S2, highlighted in blue), as compared to the control group. Histological examination of the organs collected from all the treated and control groups did not expose any histopathological changes (Fig. S2).

In terms of the intraperitoneal (IP) administration route, all four hybrid peptides caused death and/or high physical stress when injected at a concentration of 100 mg/kg. Therefore, low graded doses were attempted until we reached the maximum dose at which no signs of stress or abnormal behavior were evident. Hybrid peptides RN7-IN10 and RN7-IN8 did not display any sign of toxicity when injected at 20 mg/kg; no death occurred in any of the treated mice up to seven days post-treatment. RN7-IN9 and RN7-IN6 were non-toxic when injected at 10 mg/kg. None of the five major organs of the treated mice revealed any significant histological abnormality, as compared to the untreated control group (Fig. S3). However, mice treated with RN7-IN9 (10 mg/kg) via IP route showed significantly lower lymphocytes (p = 0.0445) and lower ALP (p = 0.0187) (Table S3, highlighted in yellow), while RN7-IN6 treated mice had lower ALT (p = 0.0425) when compared to the control group (Table S3, highlighted in blue).

In vivo antibacterial efficacy of hybrid peptides

Two peptides, RN7-IN10 and RN7-IN8, which showed the fastest killing kinetics (Jindal et al., 2015) and exhibited less toxic effects in vivo were selected to evaluate their in vivo antibacterial efficacy via IP route. Both hybrid peptides were tested at three different doses (5 mg/kg, 10 mg/kg and 20 mg/kg) in three treatment regimens (1 h, 12 h and 24 h post-infection). In the pneumococcal bacteremia model, both RN7-IN10 and RN7-IN8 failed to treat any of the infected mice at 5 mg/kg. At a dose of 10 mg/kg, 10% of the infected mice survived after treatment with RN7-IN10 (p = 0.0018), whereas 30% of the mice was able to survive after treatment with RN7-IN8 (p = 0.0002). However, at a dose of 20 mg/kg, 30% of the mice treated with RN7-IN10 survived (p < 0.0001), while 50% of the mice survived when treated with hybrid peptide RN7-IN8 (p = 0.0002) (Fig. 5). No pneumococci were detected from the blood of the mice that survived and none of the mice showed presentation of illness, as compared to the untreated group which was severely ill and inactive. Treatment via SC and IN routes had no impact on infected mice up to seven days post-infection and none of the mice survived.

Figure 5 Survival curve of infected mice treated with RN7-IN10 and RN7-IN8.

Kaplan–Meier with log-rank test (Mantel-Cox) was used to perform the statistical analysis for all of the treated groups and the untreated control using. Treatment with RN7-IN8 at 20 mg/kg displayed the highest survival rate of 50% up to seven days post-infection (p < 0.001).

In addition to our designed peptides, ceftriaxone, as a standard drug, was used to treat infected mice via IP route at 5 mg/kg, 10 mg/kg, 20 mg/kg, 40 mg/kg, and 80 mg/kg and the survival was 10%, 30%, 40%, 70% and 90% up to seven days post-infection (p < 0.0001) (Fig. 6). In the pneumococcal pneumonia model, none of the mice treated with both peptides via IP, SC and IN routes survived up to seven days post-infection and therefore, this model was excluded from further studies.

Figure 6 Survival curve of infected mice treated with ceftriaxone (CTX).

Kaplan–Meier with log-rank test (Mantel-Cox) was used to perform the statistical analysis for all of the treated group and the untreated control.

Combinations of peptide-peptide were also assessed for their ability to treat infected mice with pneumococcal bacteremia. Two combinations were used, 5 mg/kg + 5 mg/kg and 10 mg/kg + 10 mg/kg to treat mice via IP route at three regimens 1 h, 12 h and 24 h. The results indicate that the combination of 5 mg/kg + 5 mg/kg was able to treat 40% of the mice and protected them from death up to seven days post-infection (p = 0.0003), while increasing the dose to 10 mg/kg of each peptide resulted in 60% of the infected mice surviving the pneumococcal infection (p < 0.001) (Fig. 7). These findings are in agreement with our in vitro synergism results which showed that hybrid peptides are able to enhance the biological activity of each other.

Figure 7 Survival curve of infected mice treated with combinations of RN7-IN10 and RN7-IN8.

Kaplan–Meier with log-rank test (Mantel-Cox) was used to perform the statistical analysis for all of the treated group versus the untreated control.

In vivo synergy assessment of hybrid peptide RN7-IN8 in combination with ceftriaxone

Among the hybrid peptides RN7-IN10 and RN7-IN8, standalone treatment with RN7-IN8 at 20 mg/kg was found to confer significant survivability on mice infected by a highly virulent pneumococcal clinical isolate via IP route. To assess the synergistic effect of RN7-IN8 in combination with the standard drug ceftriaxone (CTX), three different doses of RN7-IN8 (5 mg/kg, 10 mg/kg and 20 mg/kg) and ceftriaxone (5 mg/kg, 10 mg/kg and 20 mg/kg) were tested, using the same bacteremia infection model in three treatment formulations: RN7-IN85–CTX5(5 mg/kg of RN7-IN8 and 5 mg/kg of CTX), RN7-IN810–CTX10(10 mg/kg of RN7-IN8 and 10 mg/kg of CTX) and RN7-IN820–CTX20(20 mg/kg of RN7-IN8 and 20 mg/kg of CTX). Using groups of 10 mice, the combinations of RN7-IN8 and ceftriaxone RN7-IN85–CTX5(5 mg/kg of RN7-IN8 and 5 mg/kg of CTX), RN7-IN810–CTX10(10 mg/kg of RN7-IN8 and 10 mg/kg of CTX) and RN7-IN820–CTX20(20 mg/kg of RN7-IN8 and 20 mg/kg of CTX) led to survival rates of 60%, 80% and 100% in mice infected with highly virulent pneumococcal strain up to seven days post-infection (p < 0.0001) (Fig. 8). Our results displayed that treatment using combinations of peptide-antibiotics conferred higher survival rate than peptide and antibiotic in their stand-alone form. In addition, all treated mice which survived from the infection appeared physically active and none of them showed signs of abnormal behavior.

Figure 8 Survival curve of infected mice treated with combinations of RN7-IN8 and ceftriaxone (CTX).

Kaplan–Meier with log-rank test (Mantel-Cox) was used to perform the statistical analysis for all treated group and the untreated control. Combination of RN7-IN and ceftriaxone at (20 mg/kg – 20 mg/kg) showed 100% survival (P < 0.0001).

Histopathological evaluation

All the histopathological examinations of the mice infected with bacteremia model with and without treatment are presented in Figs. 9 and 10. Out of the five major organs examined, the lung and spleen of the infected animals were the most severely affected. A number of histopathological changes were observed in the lung of the infected mice. As compared to the uninfected control group, the lung of the infected and untreated group exhibited extensive vascular congestion with foci consolidation. Heavy permeation of the red blood cells into the alveolar spaces strongly denoted pulmonary hemorrhage (Fig. 9A, arrow a). The greatly congested lung appeared with little alveolar spaces (Fig. 9A, arrow b). This is in contrast to the uninfected group, where the normal lung displayed greatly aerated alveolar spaces with a thin layer of the alveolar wall (Fig. 9B). Severe tissue injuries was also noticed in the spleen of the infected group (Fig. 10A). Unlike the normal spleen which showed normal red and white pulps (Fig. 10B, arrow b), the infected spleen demonstrated depleted splenocytes with no white matter/germinal center (Fig. 10A, arrow a). No significant histopathological lesions were observed in other organs, such as the brain, liver, kidney and heart.

Figure 9 Histology of lungs harvested from mice infected with S. pneumoniae receiving treatments.

(A) infected mice, (B) uninfected mice (control), (C) mice treated with RN7-IN8 (20 mg/kg), (D) mice treated with combination of RN7-IN10 and RN7-IN8 (10 mg/kg + 10 mg/kg), (E) mice treated with combination of RN7-IN8 and CTX (5 mg/kg + 5 mg/kg), (F) mice treated with combination of RN7-IN8 and CTX (10 mg/kg + 10 mg/kg), (G) mice treated with combination of RN7-IN8 and CTX (20 mg/kg + 20 mg/kg). Hematoxylin and eosin stain. Bar indicates 500 µM.

Figure 10 Histology of spleens harvested from mice infected with S. pneumoniae receiving treatments.

(A) infected mice, (B) uninfected mice (control), (C) mice treated with RN7-IN8 (20 mg/kg), (D) mice treated with combination of RN7-IN10 and RN7-IN8 (10 mg/kg + 10 mg/kg), (E) mice treated with combination of RN7-IN8 and CTX (5 mg/kg + 5 mg/kg), (F) mice treated with combination of RN7-IN8 and CTX (10 mg/kg + 10 mg/kg), (G) mice treated with combination of RN7-IN8 and CTX (20 mg/kg + 20 mg/kg). Hematoxylin and eosin stain. Bar indicates 500 µM.

For the respective treatments of infected mice including hybrid peptide RN7-IN8 at 20 mg/kg, combination of hybrid peptides RN7-IN10 and RN7-IN8 (10 mg/kg + 10 mg/kg) and combination of RN7-IN8 and ceftriaxone (5 mg/kg + 5 mg/kg, 10 mg/kg + 10 mg/kg and 20 mg/kg + 20 mg/kg), it was noticed that although lesions, inflammatory events and the degree of tissues damage were found in the organs, the degree and severity of the damage were significantly less than the infected control group. Unlike the lung of the untreated mice which exhibited severe inflammation and the alveolar spaces were about 90% congested (Fig. 9A), all the lungs harvested from the treated mice revealed only low level of congestion and minor thickening of the alveolar wall, even though these histological changes were still noticeable in the mice (Figs. 9C–9G). Treatment of infected mice with a combination of hybrid peptide RN7-IN8 and ceftriaxone at three different dosages (5 mg/kg – 5 mg/kg, 10 mg/kg – 10 mg/kg and 20 mg/kg – 20 mg/kg) showed gradual decrease in the degree of congestion and damage (Figs. 9E–9G). Lungs harvested from mice treated with a combination of RN7-IN8 and ceftriaxone at 20 mg/kg + 20 mg/kg (Fig. 9G) which presented 100% mice survival, were similar to those harvested from the uninfected control mice (Fig. 9B) in degree of normality. Likewise, all the spleens of treated mice displayed no or minimum damage, with the white and red pulps being clearly observed (Figs. 10C–10G), as compared to the infected one (Fig. 10A). No significant tissue damage was observed in the brain, kidney and liver in both treated and untreated mice.

Discussion

We report here the mechanisms of actions, in vivo toxicity and antibacterial efficiency of five hybrid peptides designed earlier in our lab, based on two templates, indolicidin and ranalexin. TEM and SEM were used to evaluate the morphological alterations caused by hybrid peptides. Results obtained from TEM displayed strong evidence that targeting the bacterial cell wall/plasma membrane is the main antibacterial mechanism used by hybrid peptides. Unlike the untreated cells, pneumococcal cells treated with hybrid peptides faced dramatic morphological changes. The breakage and fragments loss of the bacterial cell wall/membrane is probably a result of the strong interaction between the negatively charged membrane and the hybrid peptides due to their positive charge and high hydrophobic content (Jindal et al., 2015). Unlike the normal mammalian cell membranes, bacterial membranes are richer in highly electronegative lipids such as phosphatidylserine (PS), cardiolipin (CL) or phosphatidylglycerol (PG). These acidic phospholipids tend to make the bacterial membrane highly negative in charge and thus attract the positively charged antimicrobial peptides to attach to the bacterial membranes and make them preferred by AMPs over mammalian membranes (Ghavami et al., 2008). To the contrary, the membrane of mammalian cells is enriched with zwitterionic phospholipids such as sphingomyelin (SM), phosphatidylethanolamine (PE) or phosphatidylcholine (PC), which are neutral in net charge. These substances prevent the amalgamation of peptide molecules into cell membranes and thus prevent pores formation (Yeaman & Yount, 2003). In addition to their positive charge, these five hybrid peptides have a high content of hydrophobic residues. Peptide hydrophobicity is another critical property that governs the attraction of AMPs toward bacterial membrane, as it directs the level to which an AMP can penetrate into the lipid bilayer (Yeaman & Yount, 2003). The high content of tryptophan (Trp) is another advantage of these hybrids. It is well known that Trp has a significant role in the interaction of antimicrobial peptides with the bacterial membrane, as this amino acid strongly prefer the interfacial regions of lipid bilayers. In certain cases, Trp is considered hydrophobic due to its uncharged sidechain. However, it is observed that Trp residues do not reside in the hydrocarbon region of lipid bilayers and accordingly it is placed towards the more hydrophilic side of the scale (Chan, Prenner & Vogel, 2006). Another key factor of this amino acid is its ability to form an extensive π–electron system. Cation–π interaction occurs between the cationic sidechains of the basic amino acids arginine (Arg) or lysine (Lys) and the aromatic sidechains of the aromatic amino acids tryptophan (Trp), tyrosine (Tyr) or phenylalanine (Phe) (Gallivan & Dougherty, 1999). Cation–π interactions are significant for peptide self-association inside membranes and enable deeper insert into membranes by sheltering the cationic side chains (Torcato et al., 2013). The detachment of the cytoplasmic membrane from cell wall observed in pneumococcal cells upon treatment with hybrid peptides, is a possible indication of the capability of hybrid peptides to interpolate themselves between pyrophosphate-linked cell-wall anchors and the cell membrane. Subsequently, this act would pullout the isoprenyl anchor chains away from the cell membrane and weaken cell-wall adhesion. The results obtained by TEM were similar to those reported on the mechanism of nisin against B. subtilis (Hyde et al., 2006) and E. faecalis (Tong et al., 2014). Another possible explanation is that the breakage of the cell wall allows the insertion of water from the medium into the space between the two membranes and detach them (López-Expósito, Amigo & Recio, 2008). Likewise, SEM studies showed the damaging effects of hybrid peptides on the bacterial surface. Unlike the untreated cells which appeared with normal and smooth cell surface, pneumococcal cells incubated with hybrid peptides were appeared with swelling and aggregation. Besides, the numerous fragments observed in S. pneumoniae cultures treated with hybrid peptides point to a cell wall breakage and cell lysis. Membrane disruption could be associated with leakage of ions and metabolites, depolarization and eventually cell death. Adenosine triphosphate (ATP) is one of most significant molecules for all living cells, as it used as an intracellular source of energy for many biological processes (Mempin et al., 2013). In normal conditions, bacterial membranes are impervious to the efflux of ATP and other intracellular constituents, as membrane destabilization might lead to the release of normally impervious substances. Our results revealed that the ATP efflux was not increased by the incubation of pneumococcal cells with standard drugs ceftriaxone and erythromycin, but was increased by incubation with hybrid peptides. Although ceftriaxone is a member of the β-lactam family of antibiotics, its ability of releasing intracellular ATP was less than that of the hybrid peptides. This is probably due to the fact that AMPs exert their antimicrobial activity faster than standard drugs. As we have shown in our previous paper, the hybrid peptides were able to exert their bactericidal activity within one hour of incubation with resistant S. pneumoniae, whereas ceftriaxone could not eliminate S. pneumoniae up to 240 min of incubation (Jindal et al., 2015). The ATP efflux results suggest that our positively charged hybrid peptides have strong affinity to bind to the negatively charged bacterial membrane, disrupting its integrity and allowing a significant amount of ATP to be released to the surrounding environment. However, a reduction in the amount of ATP released to the medium was noticed after 1 hr of incubation; this might be due to the rapid degradation of ATP by enzymes released to the medium as a result of membrane damage, which subsequently leads to rapid cell death. Such results were also observed when Candida albicans was treated with CATH-2 peptide; the levels of ATP released after 5 min of incubation were higher than the levels of ATP after 1 hr of incubation (Ordonez et al., 2014). Altogether, results from the TEM, SEM and ATP release assay indicate that the hybrid peptides destabilize the cell envelope of the pneumococcal cells. Hence, it can be hypothesized that the disturbance of the bacterial surface must activate an autolytic and/or cell death mechanism.

It is well known that the cell membrane is not the only target for antimicrobial peptides. AMPs may also attack other cell components such as DNA, RNA or proteins (Li et al., 2013). For instance, the antimicrobial peptide buforin II has the ability to translocate itself to the inner leaflet of the plasma membrane and target the DNA after breaching the membranes, resulting in rapid cell death (Park, Kim & Kim, 1998). Like their parent peptide indolicidin which has been shown to target and inhibit DNA synthesis as one of its mechanisms of action (Subbalakshmi & Sitaran, 1998; Marchand et al., 2006), all the hybrid peptides were capable of binding to DNA efficiently and preventing it from moving down through the agarose gel. These results suggest that hybrid peptides could possess another mechanism of bacterial killing, by inhibiting intracellular functions via interference with DNA function. Hsu and co-workers have revealed through their work that the ability of the parent peptide indolicidin to permeabilize bacterial membranes is not the only mechanism of antimicrobial action. Indolicidin is also capable of binding efficiently to DNA and form a complex. The ability of indolicidin to penetrate the cell membrane allows the peptide to translocate itself to the cytoplasm and bind to the negatively charged DNA via its positive charge (Hsu et al., 2005). Moreover, Ghosh and co-workers have identified the central motif (PWWP) of Indolicidin responsible of stabilizing the DNA duplex and thus inhibiting DNA replication and transcription. The two tryptophan residues of the central motif play a significant role in stabilization of the duplex by desolvating the core of the DNA (Ghosh et al., 2014). This motif is conserved in our hybrid peptides and therefore, we hypothesize that the hybrid peptides like their parent indolicidin are most probably able to bind to bacterial DNA and preventing its intracellular function. The interaction of peptides with bacterial DNA can prevent or hinder gene expression, which is an efficient way to suppress and inhibit normal enzyme and receptor synthesis, damaging the intracellular components required for the life cycle of the bacterial cell and thus leads to cell death.

Combinations of antimicrobial agents are often used to combat multi-drug resistant isolates (Novy et al., 2011). Several studies have reported synergistic effects of combinations of AMPs with standard antibiotics. The hybrid peptide LHP7 revealed a synergistic effect against a clinical isolate of methicilin-resistant S. aureus MRSA, when combined with ampicillin (Xi et al., 2013). In the present study, we utilized the chequerboard MIC technique to assess peptide-drug interaction. Our results revealed that all the five hybrid peptides exhibited synergistic effects against pneumococcal clinical isolate, when combined with each other and with conventional drugs erythromycin and ceftriaxone. One possible explanation of the synergistic effects of peptides-drug combinations is that the hybrid peptides may increase permeability by interacting with the bacterial cell wall/membrane, making it easier for conventional drugs to act on their targets. Previous reports have shown that β-lactam antibiotics like ceftriaxone exert higher antimicrobial activity, when combined with membranolytic peptides such as nisin, as these AMPs cause changes in cell morphology by forming pores, allowing antibiotics to enhance their action and produce a greater damage within the cell wall (Singh, Prabha & Rishi, 2013; Tong et al., 2014). Another possible mechanism of synergistic combinations is that the antimicrobial peptides alter the efflux pump systems, allowing intracellular antibiotics such as macrolides to act more efficiently on their intracellular targets (Ruhr & Sahl, 1985; Soren et al., 2015).

The use of two or more antibacterial drugs in combination therapy is an alternative strategy to enhance treatment outcome in a clinical setting (Caballero & Rello, 2011). This is especially valuable in patients with severe pneumococcal infections. For instance, combination antibiotic therapy with both β-lactam and macrolide had a significantly lower case—mortally rate, in comparison with a single antibiotic therapy (Mufson & Stanek, 2006). Broad-spectrum cephalosporins such as ceftriaxone are important antibiotics in the management of invasive diseases induced by penicillin-resistant pneumococci. However, the rate of pneumococcal strains resistant to ceftriaxone has increased significantly (Chiu et al., 2007). Hence, RN7-IN8, which showed significant therapeutic efficacy in the infected mice in its standalone form, was further assessed for in vivo therapeutic synergism in combination with ceftriaxone. Combination of RN7-IN8 and ceftriaxone resulted in a synergistic effect, when tested in vivo using mice infected with pneumococcal bacteremia model. The survival rates in mice treated with this combination at varying dosages increased dramatically as compared to the sum of the survival rates of standalone treatment. These findings are in agreement with our in vitro synergism results which demonstrated that hybrid peptides and ceftriaxone can act synergistically and kill pneumococci rapidly. Unlike untreated mice that died within four days after infection, 100% of mice treated with combination of RN7-IN8 and ceftriaxone (20 mg/kg – 20 mg/kg) survived at day seven post-infection. Using a combination of these two drugs at a low dose (20 mg/kg) showed better survival rate than the use of ceftriaxone alone at a high dose (80 mg/kg), thus giving another advantage to the hybrid peptide RN7-IN8 to reduce the risk of developing resistance by the bacterial pathogen. All the mice treated with RN7-IN8 and the combinations with ceftriaxone did not show sign of sickness or abnormal behavior.

Conclusion

In sum, our hybrid peptides showed promising in vitro and in vivo antibacterial activity against S. pneumoniae. The results of the in vitro and in vivo synergism tests clearly presented that the hybrids are not only potent antimicrobials in their standalone form, but also when combined with standard antibiotics, suggesting that these peptides can be used as supporting compounds to reduce the therapeutic dose of antibiotics, thus reducing potential resistance. Although RN7-IN8 showed promising therapeutic outcome, there are some limitations in its efficacy. Primarily, the peptide had no effect on the pneumonia model, where the bacterial inoculum was administered directly into the thoracic cavity to infect the lungs, while the peptide was given at distant sites, indicating that the peptide could not diffuse effectively to the site of infection. This could possibly be due to the degradation by blood or cellular components. On the other hand, the effectiveness of the peptide in the bacteremia model is probably due to both infection and treatment being carried out at the same site. Hence, AMPs have a huge potential to play a crucial role in combating resistant bacteria, either as standalone therapeutics or in combination with other drugs.

Supplemental Information

Data S1 Raw Data 1

Survival curve analysis for mice treated with hybrid peptides at different concentrations.

Click here for additional data file.

Data S2 Raw Data 2

Survival curve analysis for mice treated with ceftriaxone at different concentrations.

Click here for additional data file.

Data S3 Raw Data 3

Survival curve analysis for mice treated with combinations of hybrid peptides at different concentrations.

Click here for additional data file.

Data S4 Raw Data 4

Survival curve analysis for mice treated with combination of hybrid peptide RN7-IN8 and ceftriaxone at different concentrations.

Click here for additional data file.

Data S5 Raw Data 5

Raw data for generating ATP release graph.

Click here for additional data file.

Figure S1 Histological morphology of different organs harvested from mice treated with peptides via SC route

(A) Control group without treatment, (B) mice injected with RN7-IN10 (100 mg/kg). (C) Mice injected with RN7-IN9 (100 mg/kg). Mice injected with RN7-IN8 (100 mg/kg). Mice injected with RN7-IN6 (100 mg/kg). Magnification at 200X, H & E staining. Bar indicates 100 µm.

Click here for additional data file.

Figure S2 (A) Control group without treatment, (B) mice injected with RN7-IN10 (20 mg/kg). (C) Mice injected with RN7-IN9 (20 mg/kg). Mice injected with RN7-IN8 (20 mg/kg)

Mice injected with RN7-IN6 (20 mg/kg). Magnification at 200X, H & E staining. Bar indicates 100 µm.

Click here for additional data file.

Figure S3 Histological morphology of different organs harvested from mice treated with peptides via IP route

(A) Control group without treatment, (B) mice injected with RN7-IN10 (20 mg/kg). (C) Mice injected with RN7-IN9 (10 mg/kg). Mice injected with RN7-IN8 (20 mg/kg). Mice injected with RN7-IN6 (10 mg/kg). Magnification at 200X, H & E staining. Bar indicates 100 µm.

Click here for additional data file.

Table S1 Whole blood haematogram and serum biochemistry of mice treated with four hybrid peptides via SC route

Click here for additional data file.

Table S2 Whole blood haematogram and serum biochemistry of mice treated with four hybrid peptides via IN route

Click here for additional data file.

Table S3 Whole blood haematogram and serum biochemistry of mice treated with four hybrid peptides via IP route

Click here for additional data file.

Supplemental Information 1 Raw data for DNA assay

Click here for additional data file.

Additional Information and Declarations

Competing Interests

Author Contributions

Ethics

Data Availability

The authors declare there are no competing interests.

Hassan Mahmood Jindal performed the experiments, analyzed the data, wrote the paper, prepared figures and/or tables.

Keivan Zandi conceived and designed the experiments.

Kien Chai Ong analyzed the data, wrote the paper, reviewed drafts of the paper.

Rukumani Devi Velayuthan performed the experiments, reviewed drafts of the paper.

Sara Maisha Rasid performed the experiments.

Chandramathi Samudi Raju conceived and designed the experiments, wrote the paper, prepared figures and/or tables, reviewed drafts of the paper.

Shamala Devi Sekaran conceived and designed the experiments, analyzed the data, contributed reagents/materials/analysis tools, wrote the paper, reviewed drafts of the paper.

The following information was supplied relating to ethical approvals (i.e., approving body and any reference numbers):

All animal experimentations were conducted according to the guidelines approved by the Faculty of Medicine Institutional Animal Care and Use Committee (FOM IACUC), University of Malaya. Ethics Reference no.: 2013-07-15/MMBTR/SDS.

The following information was supplied regarding data availability:

The raw data has been supplied as a Supplemental File.

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
