# Peer review of "Mechanisms of action and in vivo antibacterial efficacy assessment of five novel hybrid peptides derived from Indolicidin and Ranalexin against Streptococcus pneumoniae"

_PeerJ, doi:10.7717/peerj.3887_

## Round 0.1 · original submission · Major Revisions

After receiving feedback from 3 independent reviewers, it is necessary to perform additional experiments to further support your findings. Among them, it is a required that you show time and dose dependent experiments for AMP activities and ATP release. Specifically, it is compulsory that you attend comments from reviewers 1 and 2.

Reviewer 1 ·

Basic reporting

... stated in 'General comments for the author'

Experimental design

... stated in 'General comments for the author'

Validity of the findings

... stated in 'General comments for the author'

Additional comments

In the manuscript #18224v1 by Jindal and colleagues “Mechanisms of action and in vivo antibacterial efficacy assessment of five novel hybrid peptides derived from Indolicidin and Ranalexin against Streptococcus pneumoniae”, authors aim to investigate the mechanisms of action and the therapeutic efficacy of the hybrid peptides. Among other things authors assessed the effects of those peptides on the integrity of the pneumococcal cell wall/membrane. They also measured by DNA retardation assay, the impact of the peptides on the integrity of, and binding to genomic DNA of those microbes. The in vivo-toxicity and antibacterial activity was evaluated on ICR male mice. Authors have found that the hybrid peptides caused significant morphological alterations in Streptococcus pneumoniae and disrupted the integrity of the cell wall/membrane. The rapid release of ATP from pneumococcal cells after one hour of incubation suggests that the antibacterial action for the hybrid peptides is based on membrane permeabilization and damage. The DNA retardation assay revealed that at 62.5μg/ml all the hybrid peptides were capable of binding and preventing the pneumococcal genomic DNA from migrating through the agarose gel. In vitro synergy was observed when pneumococcal cells were treated with combinations of hybrid peptides with each other and with conventional drugs erythromycin and ceftriaxone. The in vivo therapeutic efficacy tests revealed that the hybrid peptide RN7-IN8 could improve the survival of pneumococcal-infected mice, as 50% of the infected mice survived up to 7 days post-infection. In vivo antibacterial efficacy of the hybrid peptide RN7-IN8 was significantly improved when combined with the standard antibiotic ceftriaxone. The presented results suggest that breaching the cell wall/membrane is likely the principal mechanism of action of the hybrid peptides. Furthermore, the hybrid peptides may affect DNA synthesis and gene expression. This is potentially-interesting manuscript, however it requires a lot of improvements.


Improvements suggestions:

1.)
Manuscript’s English language is so bad that often it is difficult to understand the meaning of the sentences (even in the summary). The manuscript MUST BE edited by a native English scientist prior to resubmission.

2.)
Please state in the figure legends 3 and 4, from how many experiment-repetitions the data has been obtained.

3.)
The data displayed in figure 3 is poorly presented. It should be rearranged and preferably split into 2 subfigures (with controls included in both), so that the ATP-release is seen as time kinetics.

4.)
The discussion needs to be expanded, especially regarding possible molecular mechanism(s) of the interactions of the dipeptides with bacterial membranes. Authors may get inspired by recent papers by Ghavami et al (doi: 10.1111/j.1582-4934.2008.00129.x), and Savelyeva et al., (doi: 10.1007/978-1-4471-6458-6_10).

Reviewer 2 ·

Basic reporting

Overall, the English is acceptable. However, the first sentence shows an unsual structure.

Reference style needs to be optimized (capitals in journal titles)

The authors use hybrid AMPs but do not explain what sort of hybrids they used; a reference to a previous paper is not enough information for a reader.

Introduction and Discussion need to be optimized; do not repeat information in both parts.

Experimental design

Quality of EM pictures is low and not very informative.

The effect of AMPs is time and dose dependent; provide such data.

ATP release: provide time and dose dependent data

AMPs can work within minutes; proide data for 10, 20, 40 min

DNA binding: why do you assume that AMPs bind to DNA; explain your rationale; Fig 4 is not convincing; do you have a positive control for a DNA binding agent?

Kaplan Meyer curves:
Better than control would be: healthy animals without infection
Not untreated: better infected animals without antimicorobial treatment

Validity of the findings

Without dose and time dependent data, this paper is rather premature. It needs more laboratory work to be conclusive.

Additional comments

Without dose and time dependent data, this paper is rather premature. It needs more laboratory work to be conclusive.

·

Basic reporting

The authors of this report present us with an interesting study. The contribution of this paper proves significant in determining the action mechanism of new antimicrobial peptides. The investigation about new molecules with antimicrobial activities is currently under study.
Introduction to this paper is comprehensive, and it provides a clear support for both the theoretical aspects and the research objectives. Literature reference is sufficient to support the study. The article structure is clear and figures and tables are appropriately used.

Some suggestions and points that need clarify are in the revised version.

Experimental design

The article is an original study, the experiments assays are appropriately designed, however in some cases the materials and methods section requires more detail and more clarity in order to be useful to future readers.
The authors should be providing information about the reproducibility of findings. How many experiments were carry out?

Validity of the findings

No comment

Additional comments

The article is original and was appropriately discussed. The comments and suggestions to make it better are in revised version (pdf).

---

## Round 0.2 · Major Revisions

I see that some improvements were made, however, both reviewer 2 and 3 still show concerns about experiments missing controls or figures lacking clarity, therefore, it is my decision that it requires major revision.

Reviewer 1 ·

Basic reporting

.

Experimental design

.

Validity of the findings

.

Additional comments

I am satisfied with the revision.

Reviewer 2 ·

Basic reporting

My main concern and criticism has not been adressed.

Introduction and discussion are not consice enough; omit information which is not necessary to understand this paper

Experimental design

My concern addressed the quality of the experiments.
It demands that some experiments need to be repeated and proper controls need to be used.

Validity of the findings

without controls the results remain difficult to interpret

Additional comments

the quality of the figures is still very bad; colours cannot be seen.
Figures are very crowded and not well designed

·

Basic reporting

The basic report is include in general comments for the author

Experimental design

No comments

Validity of the findings

No comments

Additional comments

The second version of the paper “Mechanisms of action and in vivo antibacterial efficacy assessment of five novel hybrid peptides derived from Indolicidin and Ranalexin against Streptococcus pneumoniae” was appropriately revised. However, some improvements are required.
1. Figures: The figure 2 should be described appropriately on the figure legends. In the actual figure 3 the authors should be include standard deviation and explain carefully the data of significant differences.
2. I suggest that the authors review the use of abreviations of Units following the international system of Units. Por example: abbreviation for “hora” should be “h”. Use the space between number and unit.

---

## Round 0.3 · accepted · Accept

I congratulate you for this acceptance, and look forward to seeing news manuscripts from you submitted to PeerJ.

Reviewer 1 ·

Basic reporting

I am satisfied with the revision

Experimental design

I am satisfied with the revision

Validity of the findings

I am satisfied with the revision

Additional comments

I am satisfied with the revision